# Toxic or Not Toxic, That Is the Carbon Quantum Dot’s Question: A Comprehensive Evaluation with Zebrafish Embryo, Eleutheroembryo, and Adult Models

**DOI:** 10.3390/polym13101598

**Published:** 2021-05-15

**Authors:** Chih-Yu Chung, Yu-Ju Chen, Chia-Hui Kang, Hung-Yun Lin, Chih-Ching Huang, Pang-Hung Hsu, Han-Jia Lin

**Affiliations:** 1Department of Bioscience and Biotechnology, National Taiwan Ocean University, Keelung 20224, Taiwan; jerrych0214@gmail.com (C.-Y.C.); m9541043@gmail.com (Y.-J.C.); kang083035@gmail.com (C.-H.K.); hungyun59@gmail.com (H.-Y.L.); huanging@ntou.edu.tw (C.-C.H.); 2Center of Excellence for the Oceans, National Taiwan Ocean University, Keelung 20224, Taiwan; 3Bachelor Degree Program in Marine Biotechnology, National Taiwan Ocean University, Keelung 20224, Taiwan

**Keywords:** carbon quantum dots, nanotoxicology evaluation, zebrafish, comprehensive life cycle

## Abstract

Carbon quantum dots (CQDs) are emerging novel nanomaterials with a wide range of applications and high biocompatibility. However, there is a lack of in-depth research on whether CQDs can cause acute or long-term adverse reactions in aquatic organisms. In this study, two different types of CQDs prepared by ammonia citrate and spermidine, namely CQD_AC_ and CQD_Spd_, were used to evaluate their biocompatibilities. In the fish embryo acute toxicity test (FET), the LD50 of CQD_AC_ and CQD_Spd_ was about 500 and 100 ppm. During the stage of eleutheroembryo, the LD50 decreased to 340 and 55 ppm, respectively. However, both CQDs were quickly eliminated from embryo and eleutheroembryo, indicating a lack of bioaccumulation. Long-term accumulation of CQDs was also performed in this study, and adult zebrafish showed no adverse effects in 12 weeks. In addition, there was no difference in the hatchability and deformity rates of offspring produced by adult zebrafish, regardless of whether they were fed CQDs or not. The results showed that both CQD_AC_ and CQD_Spd_ have low toxicity and bioaccumulation to zebrafish. Moreover, the toxicity assay developed in this study provides a comprehensive platform to assess the impacts of CQDs on aquatic organisms in the future.

## 1. Introduction

Engineered nanomaterials (ENMs) are tiny substances, between 1 and 100 nm, with unique optical, electrical, and chemical properties. Many ENMs have been applied in daily-use commodities. For example, silver nanoparticles are widely used as disinfectants [1,2,3]. With the widespread use of ENMs, they are inevitably released into the environment, where they may cause environmental pollution. For this reason, the toxicity of nanomaterials has received attention in recent years [4,5].

When accumulated in cells, metallic ENMs can generate free radicals, which cause DNA damage, inflammation, and oxidative stress [6,7]. For this reason, the potential toxicity of many metallic ENMs, such as those of cobalt, silver, and copper, and metal oxide nanoparticles is a concern [8,9,10]. Therefore, ENMs with low toxicity and a low environmental risk have been developed in recent years [11,12]. Among them, carbon quantum dots (CQDs), a class of emerging carbonaceous ENMs, have attracted considerable attention [13], as they can be easily synthesized and modified, are highly biocompatible, and can be used for many different purposes [14,15]. For example, using a simple dry-heating process, CQD_AC_, a negatively charged CQDs, can be synthesized from ammonium citrate and used as a fluorescent probe [16,17]. Additionally, spermidine (Spd) can be used to yield a different type of CQDs, CQD_Spd_, which exhibits a strong positive charge on its surface and excellent bactericidal activity [17,18]. Recently, a variety of CQDs have been developed using different precursors and various synthesis methods [19,20,21], and CQDs have even been purified from several kinds of processed foods consumed daily [22,23,24,25,26,27,28,29].

Although CQDs are generally considered to be highly biocompatible and safe ENMs [30], their potential toxicity is still an issue worth investigating before CQDs are widely used and mass produced. Most studies have performed experiments at the cellular level to test different types of CQDs, and showed that CQDs are less cytotoxic and more biocompatible than metallic ENMs [4,5,6]. Some studies have used animal models to evaluate the toxicity of CQDs. For example, two studies showed that a high dose (2 g·kg^−1^) of CQDs from beer or cola, administered orally to mice, results in no deaths, with major organs and levels of biochemical indicators remaining normal [27,28]. In another study, CQDs synthesized from coffee powder were fed to guppies at extremely high doses, and the results demonstrated their safety in fish [22]. 

Zebrafish and humans share more than 10,000 genes, accounting for approximately 70% of the human genome [31]. For this reason, many human drugs elicit similar physiological responses in zebrafish, making them a good model for toxicity assessments [32]. Additionally, low operation fee is another advantage, since zebrafish can reproduce a lot of offspring easily. Therefore, the Organization for Economic Cooperation and Development (OECD) suggests the use of zebrafish embryos for acute toxicity testing (OECD #236) [33]. High doses of CQDs (1.5 or 2.5 g·L^−1^) synthesized from fruits (e.g., kiwi, pear, and avocado) have been administered to zebrafish embryos, and they did not result in increased embryonic mortality or delayed hatching and development [26]. Kang et al. monitored the absorption, distribution, metabolism, and excretion of CQDs in zebrafish embryos using fluorescence, and observed that CQDs do not accumulate in zebrafish embryos nor do they interfere with their development [34]. 

Although many studies on the toxicity of CQDs have been performed, certain questions remain. First, only one type of CQDs was tested at a time in most studies, and each study might have applied different experimental conditions. As there are no standard assays to investigate CQDs toxicity, it is difficult to compare results among CQDs and studies. Second, most studies focused on the acute toxicity of CQDs, but there are almost no data related to the long-term effects of CQDs on aquatic animals.

In this study, we established a more comprehensive model to evaluate the impact of CQDs on aquatic animals. Using zebrafish embryos, eleutheroembryos, and adults, tests were designed to investigate the acute toxicity, oral subacute/subchronic toxicities, bioaccumulation, impact on fertility, and teratogenicity of two types of CQDs, negatively charged CQD_AC_ and positively charged CQD_Spd_. These tests revealed the subtle influences of CQDs on zebrafish and provide valuable reference information for future CQD evaluation.

## 2. Materials and Methods

### 2.1. Chemicals and Reagents

All general chemicals used in this study, including magnesium sulfate (MgSO_4_), calcium chloride (CaCl_2_), sodium bicarbonate (NaHCO_3_), potassium chloride (KCl), and phenylthiourea (PTU) were purchased from Sigma-Aldrich Chemical Co. (St. Louis, MO, USA). Low melting point agarose was purchased from Bionovas Biotechnology Co., Ltd. (Toronto, ON, Canada).

### 2.2. Experimental Animals

The breeding and maintenance of zebrafish (*Danio rerio*) were carried out in the same way as in a previous study [35]. In summary, the fish were maintained in aerated water at 28 °C with a fixed 14/10 h photoperiod (09:00 light-on/23:00 light-off). Each batch of embryos was obtained through random pairwise mating of five adult fish couples.

### 2.3. CQDs Preparation

Spermidine (CQD_Spd_) and ammonium citrate (CQD_AC_) CQDs were gifted by Chih-Ching Huang, National Taiwan Ocean University (Appendix A). The synthesis processes were described in previous studies [16,17,18].

### 2.4. Assessment of the Acute Toxicity of CQDs on Zebrafish Embryos and Eleutheroembryo

Following the OECD#236 test method [33], the test medium, called reconstituted water, was prepared as a solution that simulates the aquatic environment (3.5 mM MgSO_4_, 13.5 mM CaCl_2_, 3.5 mM NaHCO_3_, 50 mM KCl; pH 6.5−8.5), and the solution was aerated to oxygen saturation prior to testing. Both CQDs were diluted using the reconstituted water to 5, 10, 100, 200, and 500 ppm.

Before the experiment, single-cell stage zebrafish embryos (0.5 hpf) or eleutheroembryos (96 hpf) were washed with the test medium and randomly distributed into the wells of a 24-well plate (20 embryos/eleutheroembryos per well). Subsequently, 2 mL of CQDs solutions at different concentrations were added to the wells and a continuous 96-h apical observation was performed. During this period, the fatality rate was recorded, dead embryos were removed, and the CQDs solutions was changed every 24 h. Apical phenomena, such as embryo coagulation, lack of somite formation, non-detachment of the tail from the yolk sac, and absence of heartbeat, were used to determine embryo death. Similarly, observation of opaque milky patches on the body of fish and the absence of a heartbeat were used to determine the death of eleutheroembryos.

### 2.5. CQDs Fluorescence Distribution in Embryos and Eleutheroembryos Stages of Zebrafish

To block pigmentation and mediate the visualization of the fluorescence signals, 0.003% phenylthiourea (PTU) was added to the culture environment (test medium/CQDs solution) after 12 hpf to inhibit melanization.

For the observation of fluorescence distributions in embryos, 20 embryos (0.5 hpf) were soaked in 100 ppm CQDs for 3 h to allow for CQDs penetration. The embryos were rinsed 3 times with the test medium (reconstituted water), and then cultured in the test medium for 72 h to enable the observation of the distribution of residual CQDs during the following period. Microscopic fluorescence observations were conducted at 3, 24, 48, and 72 hpf. Similar measures applied for eleutheroembryo fluorescence distribution observations; PTU-treated eleutheroembryos (96 hpf; *n* = 20) were soaked in 2 mL of 5 ppm CQDs for 24 h. After rinsing 3 times with the test medium, eleutheroembryos were cultured in the test medium for 72 h continuously. Fluorescence microscopy observations were conducted at the 120 and 192 hpf growth stages. 

Before performing fluorescence microscopy, the organisms were rinsed with deionized water and fixed with 1.5% low-melting point agarose. The fluorescence images were taken using a confocal microscope system (C2 plus, Nikon, Tokyo, Japan).

### 2.6. Preparation of CQDs Fodders

This study adopted a method of CQDs fodders preparation based on a previous study [36]. In summary, the two CQDs were separately dissolved in deionized water (10 ppm, 2 mL), and sprayed uniformly onto 20 g of commercial fodders (final concentration 1 ppm). This mixture was incubated for 15 min at room temperature to ensure that the CQDs were properly absorbed by the feed to prevent them from being dispersed into the water during feeding. The feeds were then dried in an oven at 60 °C for 24 h. These CQDs fodders were stored at room temperature to be used for long-term weight monitoring experiments.

### 2.7. Long-Term Weight Monitoring of Adult Zebrafish Fed with CQDs Fodders

All adult fish used for the weight monitoring test were fed with commercial fodders and had reached sexual maturity before the trials. Five male and five female adult zebrafish were raised in a 1.5 L aquarium tank and fed twice a day; each fed with a fodder weight 1.5% their average weight for that week. Adult fish were divided into three groups, and fed with commercial (control), CQD_Spd,_ and CQD_AC_ fodders. The body weights of the fish were measured once a week during the 3-month period. Before measuring the weight, each individual zebrafish was removed from the aquarium tank and paper towels were used to absorb the water on its surface. Then, the fish were placed one by one in a weighing boat containing 5 mL of aerated water. A precise balance (XR-250SM-DR, Precisa, Dietikon, Switzerland) was used to measure their weights. This balance was calibrated every month.

### 2.8. Evaluation of the Fertility and Egg Hatch Rate of Adult Zebrafish Long-Term Fed with CQDs Fodders

Five adult fish were used for randomly pairwise mating after 3 months of long-term feeding with commercial (control), CQD_Spd,_ and CQD_AC_ fodders. The fish eggs were collected and incubated in a test medium for hatching, and the hatch rate was calculated.

### 2.9. Statistical Analysis

All experiments and analyses were carried out on “three biological replicates.” The experimental data were expressed as the mean ± standard deviation. Student’s t-test was used to determine the level association between data set means. Values of *p* < 0.05 were considered to indicate statistically significant differences.

### 2.10. Ethical Considerations

This study was reviewed and approved by the Institutional Animal Care and Use Committee, College of Life Sciences, National Taiwan Ocean University (IACUC Approval No. 102025).

## 3. Results

### 3.1. Evaluation of Acute Toxicity of CQDs Using the Zebrafish Embryo Survival Model

In the first test, a standard procedure of the “Fish Embryo Acute Toxicity Test”, recommended by the OECD were followed to test the toxicity of two different CQDs, CQD_AC_ and CQD_Spd_ [33]. At 0.5 hpf (hours post-fertilization), zebrafish embryos were continuously exposed to five different concentrations (5−500 ppm) of CQD_AC_ and CQD_Spd_, and the survival rates within 96 h were determined. 

The results of the embryo acute toxicity test showed an embryo survival rate greater than 50% even when the exposure concentration of CQD_AC_ was as high as 500 ppm; while the LC_50_ of CQD_Spd_ was 100 ppm (Figure 1 and Appendix A). These results indicate that low concentrations of CQDs do not have adverse effects on embryo survival. However, higher CQDs concentrations may affect the development of zebrafish embryos. The positively charged CQD_Spd_ had a more significant impact on embryo development than the negatively surface charged CQD_AC_.

### 3.2. Evaluation of CQDs Accumulation in Zebrafish Embryo Model

In this study, 100 ppm CQDs was used for the test of zebrafish embryos (0.5 hpf) by soaking them for 3 h. The 100-ppm concentration was determined based on the fact that the development of the embryo slightly interfered with this concentration (Appendix A). Thanks to the fluorescence characteristics of CQD_AC_ and CQD_Spd_, we could track the accumulation of these two CQDs in zebrafish embryos in real-time.

The results showed that the negatively charged CQD_AC_ exhibited a weak fluorescent signal under confocal fluorescence microscope. With 3D Z stack observation, fluorescence of CQD_AC_ were only surrounded the embryo (Figure 2). A reasonable assumption from this observation is that CQD_AC_ is not able to penetrate the barrier of the embryo. 

On the other hand, the distribution of CQD_Spd_ fluorescence signals was observed in the embryo’s perivitelline space and chorion (Figure 2). These results indicate that CQD_Spd_ can pass through the chorion and enter the perivitelline space to reach the embryo. It was speculated that excessive accumulation of nanomaterials might be one of the actors that affect zebrafish embryo development.

To evaluate the metabolism and excretion of CQDs in zebrafish embryo, the embryos were exposed to high-concentration CQDs solutions for 3 h, after which they were rinsed with the test medium and bred under normal conditions. At 24, 48, and 72 hpf, the CQDs fluorescence intensities and distributions during the embryo development stage were observed by fluorescence microscopy. The results showed that the fluorescence signals of both CQDs decreased significantly over time (Appendix A). Within this period, CQD_AC_ still could not penetrate the embryo, and only weak fluorescence signals were observed around the edge of the embryo. In contrast to CQD_AC_, CQD_Spd_ was found on the surface of the embryo’s perivitelline space and chorion. However, the fluorescence intensity became significantly weaker with time, and no fluorescence signals were observed in the tissues and organs of the embryo (Figure 3). These results suggest that during the development of zebrafish embryos, short-term exposure to CQDs does not cause long-term CQDs accumulation, and the CQDs tend to be excreted into the environment by diffusion.

### 3.3. Safety Evaluation of CQDs in the Eleutheroembryo Stage 

The OECD#236 “Fish Embryo Acute Toxicity Test” is a sensitive measure for the evaluation of acute toxicity of chemicals by monitoring zebrafish embryo until 96 hpf as a model. However, in our previous observation, some CQDs may not penetrate the barrier of embryo shell during this period of time so that the result might not reveal the actual toxicities of CQDs on zebrafishes. Therefore, the eleutheroembryo stage of zebrafish at 96 hpf was chosen as a test model because, in this period, eleutheroembryos are under the final stage of morphogenesis—they continue to grow rapidly and their skin comes into direct contact with compounds in the environment. It would be an ideal window to observe the potential impacts of CQDs on zebrafishes.

In this study, 96 hpf eleutheroembryos were exposed to dilution water containing 5 different concentrations of CQD_AC_ and CQD_Spd_ for 72 h, and the survival rates of the eleutheroembryos were recorded. The LC_50_ values for CQD_AC_ and CQD_Spd_ were 340 ppm and 55 ppm, respectively (Figure 4 and Appendix A). Both CQDs showed lower LC_50_ to zebrafish eleutheroembryos than embryos. 

The bioaccumulation of CQDs was also evaluated at the eleutheroembryo stage. In this study, 96 hpf eleutheroembryo were soaked in 5 ppm CQDs solutions. This concentration was determined based on the fact that no adverse effects were found on the survival of eleutheroembryo. After the eleutheroembryos were soaked for 24 h, the initial fluorescence distributions were recorded. Fluorescence microscopy images showed that after 24 h of exposure, both CQDs could enter the bodies of eleutheroembryos; clear fluorescence signals were observed in the yolk sac, intestines, and pancreas (Figure 5a).

Subsequently, these zebrafish eleutheroembryos were moved to freshwater. They were maintained in reconstituted water (returned to normal conditions) for 72 h for the evaluation of CQDs bioaccumulation. It is clearly seen that the fluorescence signals of both CQDs in the yolk sac significantly decreased after returning to normal conditions for 72 h. The fluorescence signals in the intestines gradually moved to the end of the intestinal tract (Figure 5b). These results imply that at an appropriate dose, even if CQDs penetrate fish’s bodies, they could be excreted through the digestive and excretory systems of eleutheroembryos.

### 3.4. Safety Evaluation of CQDs in Adult Zebrafish

For the evaluations of subacute/subchronic toxicity effects of CQDs, low dose and long-term exposure of CQDs by orally feeding of CQDs to adult fish was conducted. To evaluate these effects, we measured the changes in the body weights of adult fish, and the fertilities of the fishes over a long period feeding were also evaluated.

We prepared feeds containing the two CQDs at a final dose of 1 ppm. Twenty adult zebrafish were fed continuously with these feeds for 12 weeks. The body weights of all the fish were recorded weekly. After long-term feeding for 12 weeks, the average weight changes of the fish in the control and CQD_AC_ groups were −4% and +3.1%, respectively. The average weight of zebrafish in the CQD_Spd_ group increased by 16.3% (Figure 6); this was a significant change (Table 1, *p* < 0.01).

Furthermore, fertility was also evaluated in long-term CQDs fed adult zebrafish. Five adult zebrafish continuously fed with CQDs for 2−12 weeks were selected for mating, and the hatch rate and offspring health were observed and recorded. The results showed that long-term feeding with CQDs did not affect the fertility of zebrafish (F_0_). As with the adult zebrafish in the control group, those in the groups fed with CQD_AC_ and CQD_Spd_ could still lay eggs normally, and the fertility of F_0_ was not affected. The egg hatch rate of zebrafish offspring (F_1_) was also evaluated. The hatch rates of all groups were over 90% (Table 2). In addition, after one month of rearing, the appearances and body lengths of F_1_ zebrafish fed with CQDs were not different from those of fish in the control group, and no malformations were found (Appendix A). The results of these safety assessments show that long-term feeding with low-dose CQDs has almost no adverse effect on the development of adult zebrafish and their offspring.

## 4. Discussion

In the fish embryo acute toxicity test (FET), over 95% of zebrafish embryos survived during exposure to 100 ppm of CQD_AC_. When the dose was elevated to 500 ppm, only 40% of embryos died (Figure 1 and Appendix A). CQD_AC_ has much better biocompatibility than other metallic ENMs [4,5]. However, Dias et al. reported that exposure to 1500 ppm of CQDs, synthesized from fruits such as kiwi, pear, and avocado, does not cause toxicity in zebrafish embryos [26]. In addition, Xu et al. used citric acid as a precursor to prepare CQDs, which were not toxic to zebrafish embryos at a concentration of 1000 ppm [37]. Regarding the toxic effects of these CQDs differ among studies, it may be owing to the different assay parameters applied. As zebrafish embryos are very sensitive to chemicals and their development is very fast, a general guideline, such as OECD #236, should be followed to provide a standard protocol for CQDs toxicity comparison. In our experiments, the OECD #236 protocol was strictly followed. For example, only zebrafish embryos less than 0.5 hpf were used and validation procedures, such as analysis of overall fertilization rate, water temperature, negative and positive controls, and dissolved oxygen concentration, were conducted in parallel with the CQDs toxicity assays.

The various toxic effects of CQDs may also originate from their precursor molecules and synthetic methods [38]. Under the same experimental conditions, CQD_Spd_ was more toxic to zebrafish embryos than CQD_AC_ (Figure 1). According to our previous studies [16,17,18], the particle size (CQD_AC_ = 4.1 ± 1.20 nm; CQD_Spd_ = 6.3 ± 1.35 nm) and core composition (graphite structure) of both CQDs are similar, whereas their functional groups and surface charges are different. Unlike most negatively charged CQDs (i.e., CQD_AC_), positively charged CQD_Spd_ can destroy bacterial cell membranes [17,18,19]. Our previous study also showed that CQD_Spd_ temporarily opens the tight junctions of the epithelium [18]. By tracking the fluorescence of CQDs, CQD_Spd_ was observed to penetrate the chorion and enter the perivitelline space, whereas CQD_AC_ only attached to the outside of the eggshell (Figure 2). The tissue penetration ability of CQD_Spd_ may be the main factor why CQD_Spd_ interferes with zebrafish embryo development to a greater extent than CQD_AC_.

However, the fluorescent signals of both types of CQDs gradually disappeared in the zebrafish embryo within 72 h (Appendix A), indicating that CQDs are excluded from the embryo gradually. This result is similar to the observation of Kang et al., who used CQDs synthesized from glucose and ethylenediamine [34]. This indicates that rapid elimination may be a common feature among CQDs. Moreover, previous studies have reported that animal enzymes, such as eosinophil peroxidase and myeloperoxidase, decompose carbon nanomaterials [39]. In addition to excretion, the biodegradable properties of CQDs contribute to their rapid elimination in zebrafish embryos.

In the fish eleutheroembryo toxicity test (FEET), the LD_50_ of both CQD_AC_ and CQD_Spd_ was significantly decreased, compared with the FET test (Appendix A). This may be attributed to the lack of protection by the eggshell after hatching, which makes eleutheroembryos more sensitive to CQDs than zebrafish embryos. Previous studies have reported that chorion pore canals are blocked by carbon nanomaterials, such as graphite oxide or single-wall carbon nanotubes, which may cause defects in embryonic development. This problem might be caused by high concentrations of CQDs [40,41]. 

During normal embryonic development, after hatching, it takes approximately 120 h for zebrafish to develop protective skin [42]. After exposure to only 5 ppm of any CQDs at 96 hpf, CQDs was observed to enter the eleutheroembryo body and accumulate in the yolk sac, pancreas, and intestines. However, after returning the zebrafish to the normal culturing environment for 72 h, the CQDs-dependent fluorescent signals in the eleutheroembryo were significantly reduced and concentrated at the end of the intestinal tract (Figure 5). A similar observation was reported by Kang et al. with another type of CQDs [34]. This phenomenon implies that although CQDs enter the eleutheroembryo body, they are eventually excreted through the zebrafish alimentary canal and not accumulated in the body. 

Over 120 hpf, zebrafish have protective skin and mucus [42]; thus, they are better protected against environmental CQDs. Even if a CQDs enters the fish body, based on our results, they tend to be accumulated in the alimentary canal (Figure 5). Therefore, we believe that the oral administration of CQDs is a better approach to evaluate the health effects of CQDs on adult fish. In the 12-week trial, zebrafish were fed fodder containing 1 ppm of CQDs daily. Both CQD_AC_ and CQD_Spd_ did not affect the survival rate of zebrafish, although it is worth noting that the body weight of zebrafish fed CQD_Spd_ significantly increased (Figure 6 and Table 1). As Spd has multiple beneficial physiological effects [43], Spd may be gradually released from CQD_Spd_ and exert growth-promoting effects in zebrafish. In our previous study, CQD_Spd_ was applied as a feed additive to prevent infectious diseases in shrimps [36]. With antibiotic-like properties, CQD_Spd_ may also have antibiotic growth promotion effect, as long-term feeding of 1 ppm of CQD_Spd_ resulted in body weight increases in zebrafish [18]. No matter whether CQD_AC_ or CQD_Spd_ was added to the fodder, the embryo hatching rate during the 12-week trial was higher than 90% in all groups, and no significant adverse effects were observed (Table 2). Moreover, in the subsequent observation of these embryos, no embryonic development delay or teratogenesis was observed (Appendix A). Many reports have indicated that metallic ENMs affect the reproductive system or offspring of animals [44,45]. However, our results indicate that the long-term intake of CQDs does not affect the reproduction of zebrafish, potentially because CQDs are more easily metabolized and excreted, and are less bio-accumulative than other metallic ENMs. 

The results of this study showed that different types of CQDs have different toxic effects and elicit different physiological responses in zebrafish embryos, eleutheroembryos, and adults. FET is a sensitive and high throughput assay to evaluate the toxic effects of CQDs [33]. When higher sensitivity is required, FEET can be considered as a better approach for toxicity assessment [46]. Furthermore, an experimental guideline, such as OECD #236, should be strictly followed to provide an objective result for comparison. Our study was the first long-term evaluation of CQDs toxicity in zebrafish. Although exposure to high concentrations of CQD_Spd_ may interfere with the survival of zebrafish embryos and eleutheroembryos, no adverse effects were observed in the long-term low-dose feeding experiments. Our results indicate that CQDs are highly biocompatible owing to their low bioaccumulative property, whether in embryos, eleutheroembryos, or adults. Our comprehensive examination of CQDs toxicity in different zebrafish life cycle stages may provide a reference for the toxicity evaluation of other CQDs in the future.

## Figures and Tables

**Figure 1 polymers-13-01598-f001:**
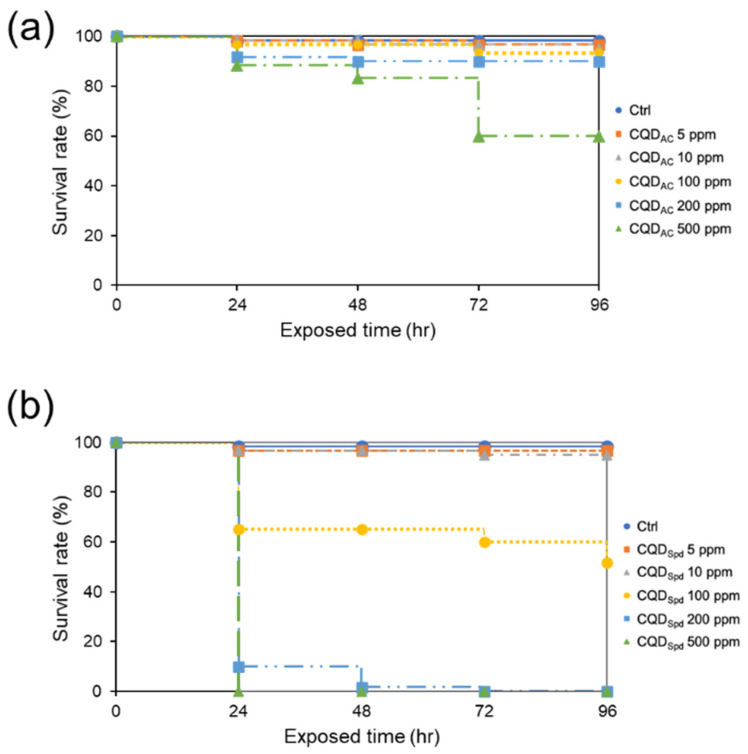
The effective concentration of CQD_AC_ and CQD_Spd_ in fish embryo acute toxicity (FET) test. The survival rate of 0.5-hpf zebrafish embryos soaked in different concentrations of (**a**) CQD_AC_, or (**b**) CQD_Spd_ solutions within 96 h (*n* = 20).

**Figure 2 polymers-13-01598-f002:**
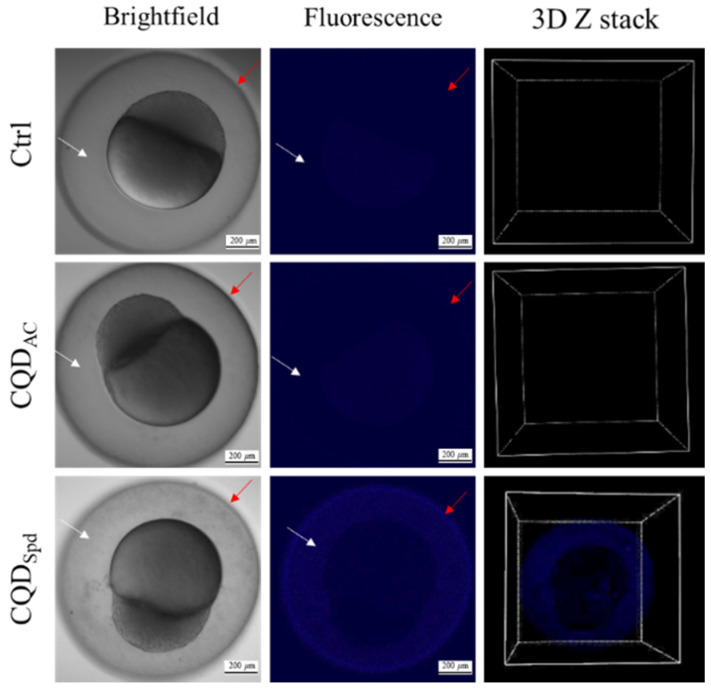
Brightfield, fluorescence, and 3D Z stack images of zebrafish embryos after soaking for 3 h in 100 ppm of CQD_AC_ or CQD_Spd_ solutions. 0.5-hpf zebrafish embryos were exposed to 100 ppm CDQs for 3 h. The distribution of the different surface charged CQDs in embryos was observed by confocal fluorescence microscopy. The white arrow indicates the perivitelline space; the red arrow represents the relative position of chorion. Scale bar = 200 μm.

**Figure 3 polymers-13-01598-f003:**
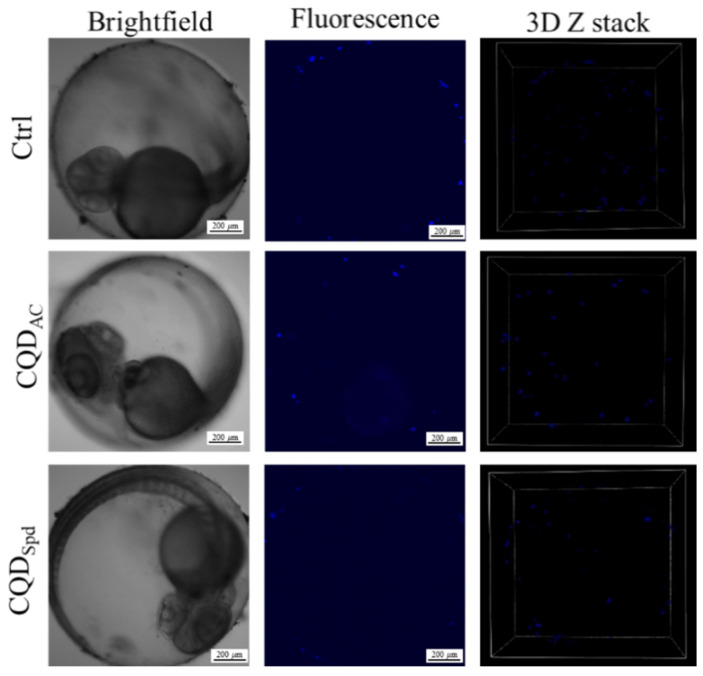
Residual fluorescence and accumulation of CQDs 72 h after return to normal conditions. After exposure to 100 ppm CQDs solutions 3 h, the embryos were rinsed and maintained in the test medium for 72 h. Residual fluorescence and accumulation of CQDs were observed by confocal fluorescence microscopy. Scale bar = 200 μm.

**Figure 4 polymers-13-01598-f004:**
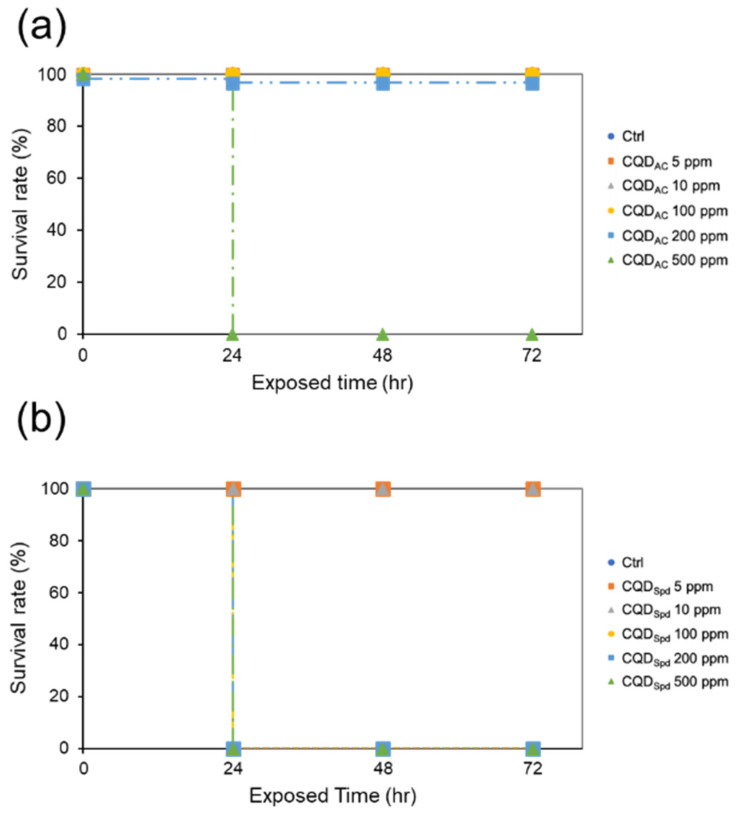
The effective concentration of CQD_AC_ and CQD_Spd_ in the fish eleutheroembryo acute toxicity (FEET) test. The survival rates of 96-hpf eleutheroembryo soaked in different concentrations of (**a**) CQD_AC_ or (**b**) CQD_Spd_ solutions within 72 h.

**Figure 5 polymers-13-01598-f005:**
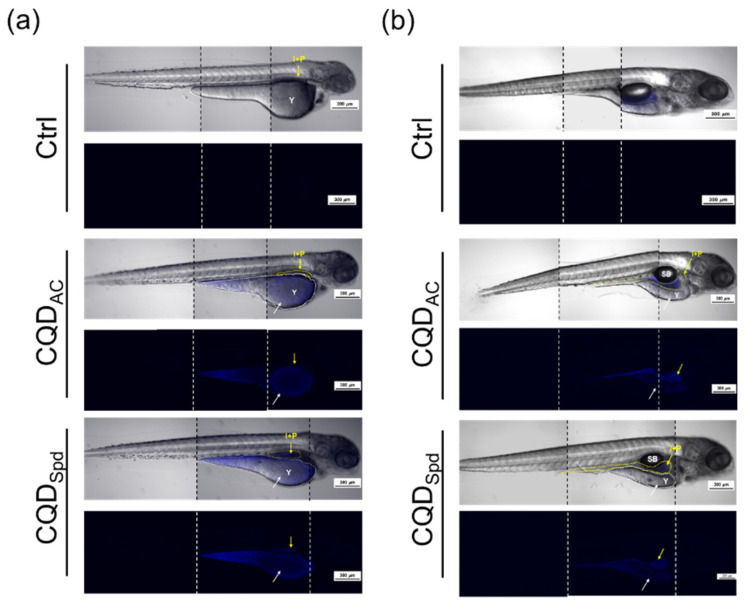
Distribution and bioaccumulation of CQD_AC_ and CQD_Spd_. Bright field and fluorescence images of zebrafish eleutheroembryo (**a**) after 24 h exposure of CQDs and (**b**) 72 h after return to normal conditions. The yellow arrow indicates the area containing the intestines (I) and the pancreas (P); the white arrow represents the location of the yolk sac (Y). Scale bar = 300 μm.

**Figure 6 polymers-13-01598-f006:**
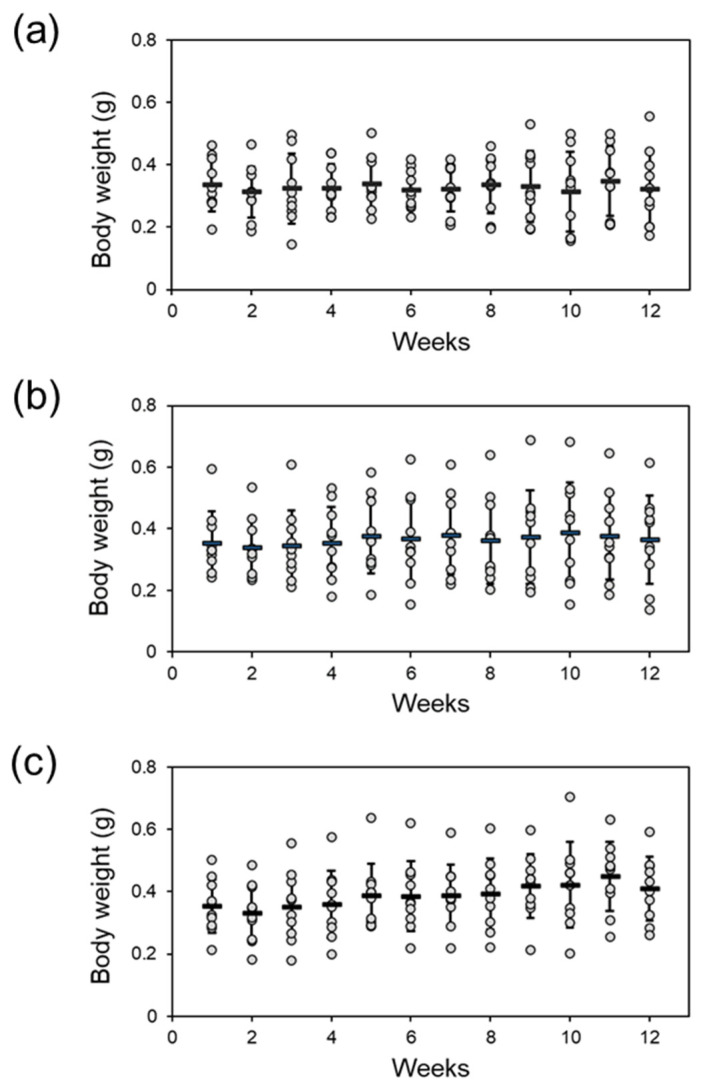
The body weight changes of adult zebrafish after feeding CQDs contained-fodders. The adult zebrafish were divided into three groups and fed with (**a**) commercial fodders (Ctrl group), (**b**) CQD_AC_ fodders or (**c**) CQD_Spd_ fodders. The body weight of each individual zebrafish was monitored weekly.

**Table 1 polymers-13-01598-t001:** Body weight comparison of adult zebrafish fed CQDs-containing fodders.

	Ctrl Group ^1^		CQD_AC_ Fodder		CQD_Spd_ Fodder	
Gender	Male	Female	Male	Female	Male	Female
0 week (g)	0.317	0.354	0.328	0.379	0.334	0.371
12 weeks (g)	0.257	0.385	0.280	0.449	0.344	0.475
Change in body weight	−18%	8.8%	−14%	18.7%	3.1%	28.2%
Average	−4.2%		3.1%		16.4%	
Statistics	−		*		*	

^1^ The adult zebrafishes in Ctrl group were fed with commercial fodders. * The asterisk indicates that the group is significantly different (*p* < 0.05) from control group using Student t-test.

**Table 2 polymers-13-01598-t002:** The hatch rates of adult zebrafish fed with CQD-containing fodders.

	Ctrl Group ^1^	CQD_AC_ Fodder	CQD_Spd_ Fodder
Week 2	98%	89%	96%
Week 4	95%	92%	91%
Week 6	97%	95%	94%
Week 8	96%	93%	97%
Week 10	90%	94%	97%
Week 12	95%	97%	96%

^1^ The adult zebrafishes in Ctrl group were fed with commercial fodders.

## Data Availability

Not applicable.

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
