# Peer review of "Toxic or Not Toxic, That Is the Carbon Quantum Dot’s Question: A Comprehensive Evaluation with Zebrafish Embryo, Eleutheroembryo, and Adult Models"

_polymers, 2021, doi:10.3390/polym13101598_

Round 1

Reviewer 1 Report

Manuscript of Lin and co-workers reports the toxicity effects of two different CQDs, obtaining from amonium citrate and spermidine, to some zebrafish samples, finding that no significant toxicity can be detected in short and long time. This manuiscript is interesting, in particular for researchers active in this field.

However, only the electrostatic nature of the CQDs has been evaluated. What is the effect of the dimensions? Some morphological information on these nanoparticles should be reported in the manuscript.

Taking into account these consideration, I suggest the publication after some revisions.

  • title is too much long. A shorter title should be more actractive
  • CQDac has been definied in the abstract, and should be defined also in the main text
  • Line 70: please, change “suggest” with “suggests”
  • if the authors do not report the synthetic procedure, section 3 can be removed,
  • Figures, Tables and Schemes must be reported in the main text, and not in a separate section
  • Data reported in Figure 1 and 4 should be shown by an instogram
  • In line 325, the authors report that “The various toxic effects of CQDs may also originate from their precursor molecules and synthetic methods”: there are some evidences?

Author Response

Reviewer#1

Point-to-point Responses to reviewers' comments:

1) However, only the electrostatic nature of the CQDs has been evaluated. What is the effect of the dimensions? Some morphological information on these nanoparticles should be reported in the manuscript.

Response: We have modified the description (Lines 337-338 in the Discussion section) and the result of this statistical analysis (Table S1) in the revised manuscript.

2) Title is too much long. A shorter title should be more attractive.

Response: Thank to the reviewer’s advice. After our discussion, we proposed 2 new titles. The first one is “Toxic or not toxic that is the carbon quantum dot’s question: A comprehensive evaluation with zebrafish embryo, eleutheroembryo and adult fish models.” The second one is “A comprehensive biotoxicity assessment of carbon quantum dots through embryo, eleutheroembryo and adult zebrafish.” We preferred the first one as the new title of our work.

3) CQDAC has been defined in the abstract, and should be defined also in the main text.

Response: We followed the suggestion and included the information in the Introduction section (Lines 47-49).

4) Line 70: please, change “suggest” with “suggests”.

Response: Your kindly suggestion was followed.

5) Figures, Tables and Schemes must be reported in the main text, and not in a separate section.

Response: Your kindly suggestion was followed.

6) Data reported in Figure 1 and 4 should be shown by an instogram

Response: We believe that the current Figure 1 and Figure 4 are better presentation format. Histogram can only represent the end point survival rate at different concentrations, thereby it will lose the information of mortality at different exposure time. For example: CQDAC exhibited low cytotoxicity in the FET experiment. Therefore, we convinced that the Kaplan-Meier method can fully demonstrate the significance of this experiment. However, we also agree with your suggestion that histogram is much easy to observe LD50. Hence, we added Fig. S1 in Supporting Information to summarize Fig 1 and 4 in histogram format.

7) In line 325, the authors report that “The various toxic effects of CQDs may also originate from their precursor molecules and synthetic methods”: there are some evidences?

Response: We have added ref 38 to support this argument.

References:

  1. Peng, Z.; Liu, X.-J.; Zhang, W.; Liu, Z.-F.; Zhang, C.; Liu, Y.; Shao, B.-B.; Liang, Q.-H.; Tang, W.-W.; Yuan, X.-Z. Advances in the application, toxicity and degradation of carbon nanomaterials in environment: A review. Int. 2020, 134, 105298.

Reviewer 2 Report

This paper describes the biocompatibility of carbon quantum dots (CQDs). In recent years, carbon quantum dots have been attracting attention for applications such as fluorescent markers that can be incorporated into living organisms. On the other hand, there is little data on the evaluation of toxicity. Recently, zebrafish have been increasingly used for toxicity evaluation from the viewpoint of ethical consideration. From this point of view, it is valuable that the toxicity of CQDs was evaluated using zebrafish in this paper. That the low toxicity is also demonstrated is important.

The experimental results are well summarized, and I think it is no problem in publishing the paper as is.

Author Response

Reviewer#2

Point-to-point Responses to reviewers' comments:

1) The experimental results are well summarized, and I think it is no problem in publishing the paper as is.

Response: Thank you very much for agreeing with our article.
